# Osteo-pathological analysis provides evidence for a survived historical ship strike in a Southern Hemisphere fin whale *(Balaenoptera physalus)*

**Hannah Viola Daume**[1,2]*, **Helena Herr**[2], **Heinrich Mallison**[3], **Matthias Glaubrecht**[1,4], **Thomas M. Kaiser**[1]

**1** LIB–Leibniz Institute for the Analysis of Biodiversity Change, Museum of Nature Hamburg, Hamburg, Germany, **2** Institute of Marine Ecosystem and Fishery Science, Universität Hamburg, Hamburg, Germany, **3** Palaeo3D, Rain, Germany, **4** Department of Biodiversity of Animals, Universität Hamburg, Hamburg, Germany

* viola.daume@gmail.com

**Data Availability Statement:** All relevant data are within the paper.

## Abstract

The life history of a fin whale *(Balaenoptera physalus)* caught during whaling operations in the 1950s was partly reconstructed. 3D surface models of the bones of the skeleton curated at the Zoological Museum of Hamburg were used for an osteopathological analysis. The skeleton revealed multiple healed fractures of ribs and a scapula. Moreover, the processus spinosi of several vertebrae were deformed and arthrosis was found. Together, the pathological findings provide evidence for large blunt trauma and secondary effects arising from it. Reconstruction of the likely cause of events suggests collision with a ship inflicting the fractures and leading to post traumatic posture damage as indicated by skeletal deformations. The injured bones had fully healed before the fin whale was killed by a whaler in the South Atlantic in 1952. This study is the first in-detail reconstruction of a historical whale—ship collision in the Southern Hemisphere, dating back to the 1940s, and the first documentation of a healed scapula fracture in a fin whale. The skeleton provides evidence for survival of a ship strike by a fin whale with severe injuries causing long-term impairment.

## Introduction

The Zoological Museum of Hamburg houses a large collection of marine mammal skeletons, of which only a few are currently on display. The fin whale *(Balaenoptera physalus)* skeleton (ZMH S 10261/10616) was brought back to Hamburg by Dr. Kurt Schubert [1] who accompanied the second whaling campaign financed by Aristoteles Onassis in 1952. Past World War II Hamburg-based whaling campaigns were financed by Greek shipowner Aristoteles Onassis using Panama and Honduran whalers to evade the ban on German whaling imposed as part of the Potsdam Conference [2,3]. Due to space limitation, the skeleton could not be mounted and displayed until in 2015 new exhibition space became available and efforts were undertaken to partly mount the skeleton for display.

**Funding:** The author(s) received no specific funding for this work.

**Competing interests:** The authors have declared that no competing interests exist.

On this occasion, we investigated the skeleton and documented several deformations and abnormalities, such as healed bone fractures, deformations of the neural spine and arthrosis. These findings called for a more thorough investigation of the skeleton in order to answer anatomical as well as osteopathological questions, ultimately in an attempt to reconstruct the life history of this particular specimen of fin whale.

Only a few attempts have been made to reconstruct the cause of traumata in whale skeletons from museum collections. Hellier, Hufthammer [4] investigated a fin whale and a humpback whale *(Megaptera novaeanglia)* skeleton collected in the 19th century, which both exhibited notable osteological pathologies. Based on visual inspection, they concluded, that the fin whale had likely suffered trauma from a ship collision. However, their investigations were restricted to visual inspection of the individual damaged bones.

Today, 3D modelling is a useful tool for surface digitising, allowing visual as well as quantitative analysis of bone specimens. The digital representation then serves as a reference of the surface morphology, while the original specimen can remain untouched. Besides of the ease in handling bulky or delicate specimens just in the virtual world, 3D models also help perfect visualisation and may serve educational purposes among many others. 3D models can then be used for e.g., for the digital assembly of a skeleton and for an animated visualisation and size-reduced 3D printouts. In this study, we used different 3D surface modelling techniques to build a virtual model of the entire fin whale skeleton and to employ those models to document its osteopathology. We then combined this virtual skeleton with a digital ship model to virtually reconstruct a plausible scenario of a ship—whale collision explaining the observed injuries and consequences.

## Materials and methods

The bones of the fin whale skeleton (ZMH S 10261/10616) were first visually inspected and arranged. We then measured and described all bones according to Buchholtz and Schur [5] and Carrillo, Alcántara [6]. Subsequently, digital surface models of each individual bone or bone fragment were created applying two different methods of 3D surface data acquisition: 1) Structured light scanning using Artec 3D scanning systems (https://www.artec3D.com) was performed by 3D Manufactur Seevetal, and 2) photogrammetry was conducted using Agisoft Photoscan (www.agisoft.com) and Meshlab [7]. The latter technique was used for the cranium and other bones that proved difficult to handle or even not suitable for Artec 3D scanning gear. All surface models were converted to stereolithography format (*.STL) and digitally mounted in Rhinoceros 5.0. [8]. For the mounting we followed the protocol of Mallison [9] in order to avoid the influence of preconceived notions on the mount. We then adjusted the skeleton based on our experience from whale dissections and aerial and subaqueous photographs of fin whales. Subsequently, the skeleton was visualised, measured in full length and the pathologies described. Based on the virtually mounted model of the skeleton a virtual scenario of a ship—whale collision was generated as the most plausible cause of the observed trauma. Given the size of fin whales, only a virtual skeleton allowed a holistic assessment of the spatial distribution of the pathologies in relation to a ship hull.

## Results

Based on the 3D model of the skeleton (Fig 1) we reconstructed a total body length of 19.54 m, indicating an adult individual [10]. 170 bones were preserved, including 62 vertebrae (C7 T14 L16 Ca25) and 14 pairs of ribs. The hyoid bone, the pelvis, the bullae and some of the digital bones were found missing.

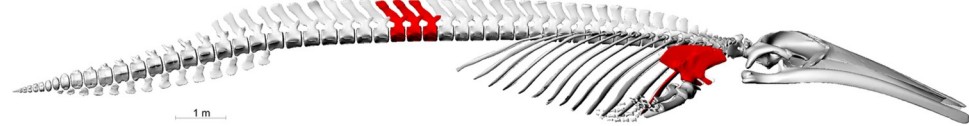

**Fig 1. 3D model of fin whale** *(Balaenoptera physalus)* **skeleton ZMH S 10261/10616).** Bones showing healed fractures are marked in red.

All epiphyses were fused with their respective vertebral bodies indicating the specimen was physically mature [11–13]. Abnormalities of different types were detected in 19 out of the 170 bones available, including healed fractures, arthrosis and deformations. The anomalies are described in detail in the following:

## Scapula

The right scapula body (Figs 2–4) exhibits a simple extraarticular fracture that descends sagittally through the entire bone. The resulting two fragments moved one upon the other and healed *ad longitudinem cum contractione* [14] resulting in an offset of ca. 10 cm (see Fig 5 for comparison with unfractured left scapula). On the distal end of the fracture line a large mass of callus tissue formed.

## Ribs

A simple extraarticular fracture is evident in the proximal sections of both the first (Fig 6) and second (Fig 7) right rib, respectively. The fractures descend transversally and separate both ribs into a short proximal and a longer distal fragment each. Significant calli formed during

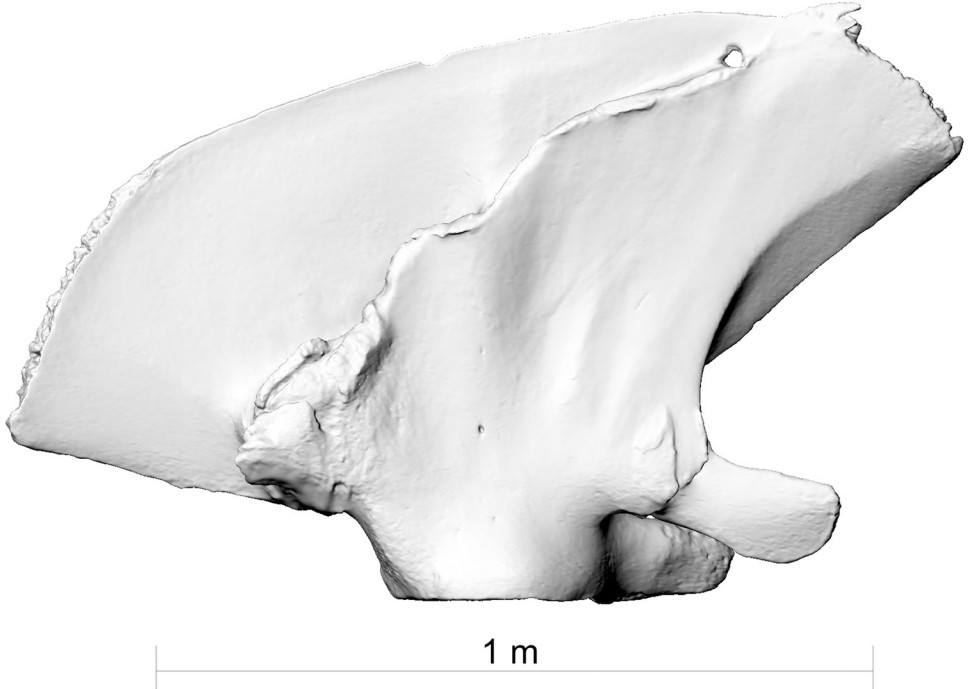

1 m

**Fig 2. The right scapula of fin whale ZMH S 10261/10616 from the external side.** Note the healed fracture.

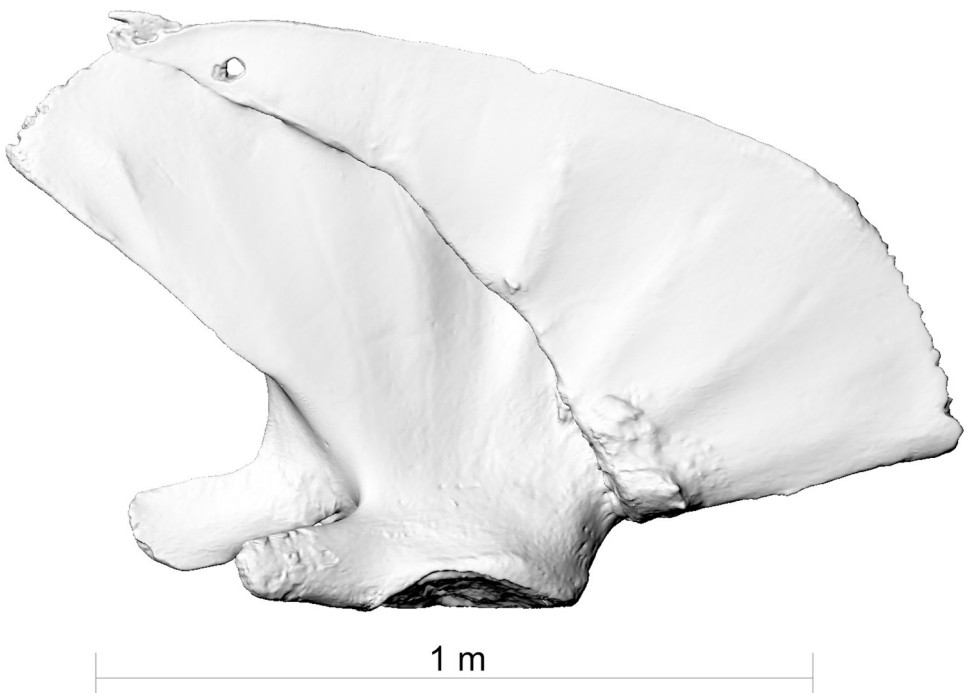

**Fig 3. The right scapula of fin whale ZMH S 10261/10616 from the internal side.** Note the healed fracture.

the healing process, encapsulating the fractures. In both ribs the fragments healed in slightly displaced positions *ad latus* [14].

## Vertebrae

Lumbar vertebrae L8, L9 and L10 (Fig 8) show fractures on their distal portion of their processus spinosi. These fractures descend sagitally through the individual processus forming a caudally descending line across the three processes (Fig 9). Lumbar vertebra L8 healed *ad latus*, lumbar vertebrae L9 and L10 healed displaced *ad longitudinem cum contractione* and thereby had shortened the processus.

Several lumbar vertebrae anterior (L5-L7) and posterior (L11-L15) of the fractured vertebrae (L8-10) show deformations in their processus spinosi (Figs 10 and 11). L5-L7 and L11-L13 are bended towards the left, L14-L15 are bended to the right (Fig 12).

Some thoracic vertebrae showed proof of arthrosis in their zygapophysial joints (Fig 13). The postzygapophysial joints were affected by zygarthrosis [15] in the thoracic vertebrae T2 and T3, both post- and praezygapophysial joints were affected in the thoracic vertebrae T4 and T5. The thoracic vertebrae T6 showed evidence of zygarthrosis in its praezygapophysial joints.

## Discussion

The skeleton shows two major areas of severe trauma that cannot be easily explained without assuming a major external force acting on the bones. A fractured scapula is a rare observation in necropsied cetaceans and cetacean skeletons, and a comparatively rare injury in vertebrates because the scapula bone is particularly sturdy [16]. In small cetaceans fractured scapulae occur mainly as a result of intra-interspecific interaction [17]. In large baleen whales, to our

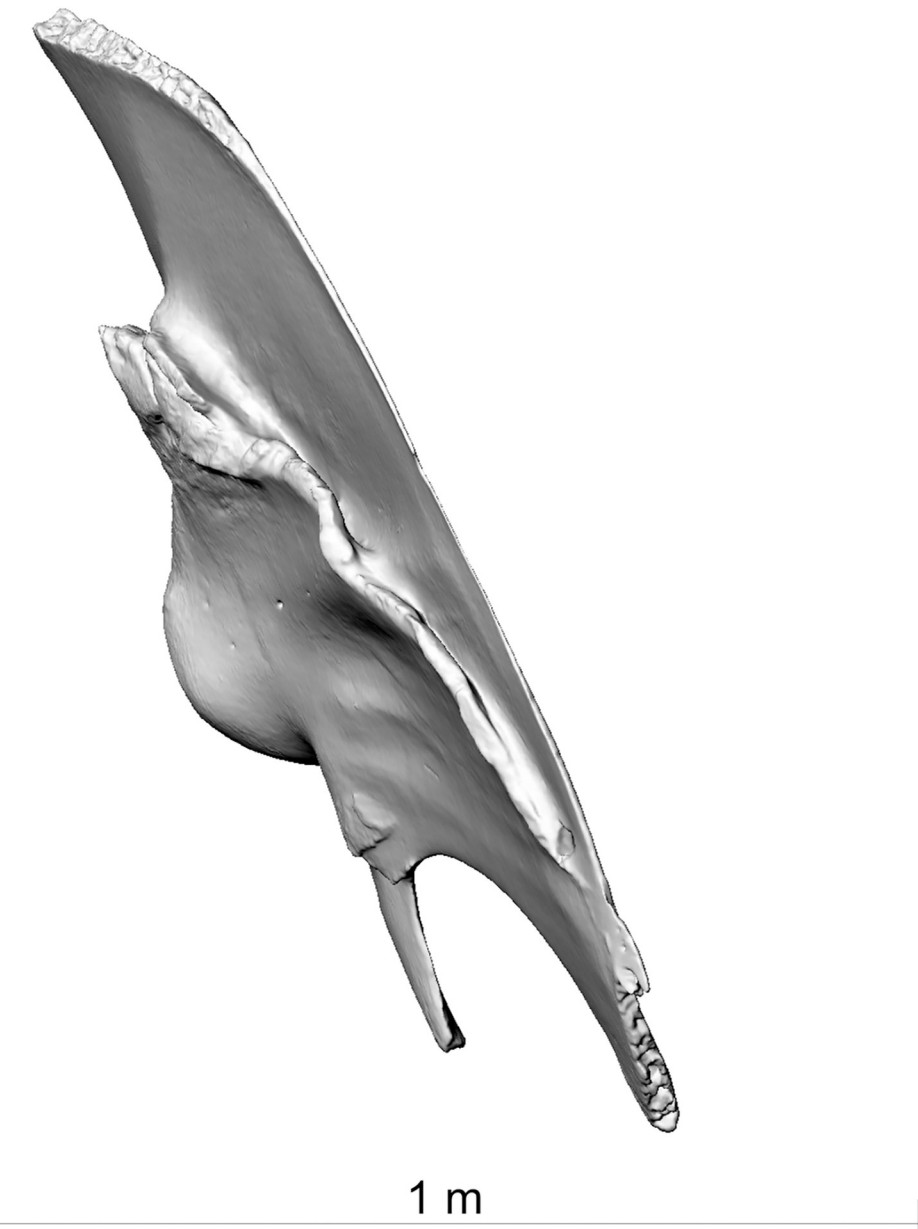

**Fig 4. The right scapula of fin whale ZMH S 10261/10616 in dorsal view.** Note the healed fracture.

knowledge only one case of a fractured scapula has been described from a necropsy of a North Atlantic right whale (*Eubalaena glacialis*, [18]). Killer whales *(Orcinus orca)* have been shown to cause rib fractures also in large baleen whales [19,20], but a fracture to the scapula, which is a much stronger and more robust bone than the ribs, is unlikely to have been caused by a killer whale attack. Therefore, we can exclude other marine organisms as the culprit.

The most plausible agent for this kind of blunt force trauma causing such fractures is a ship strike [21]. Trauma resulting from a vessel collision is characteristically marked by a premortem fracture or displacement of skeletal elements [21]. In fact, Campbell-Malone,

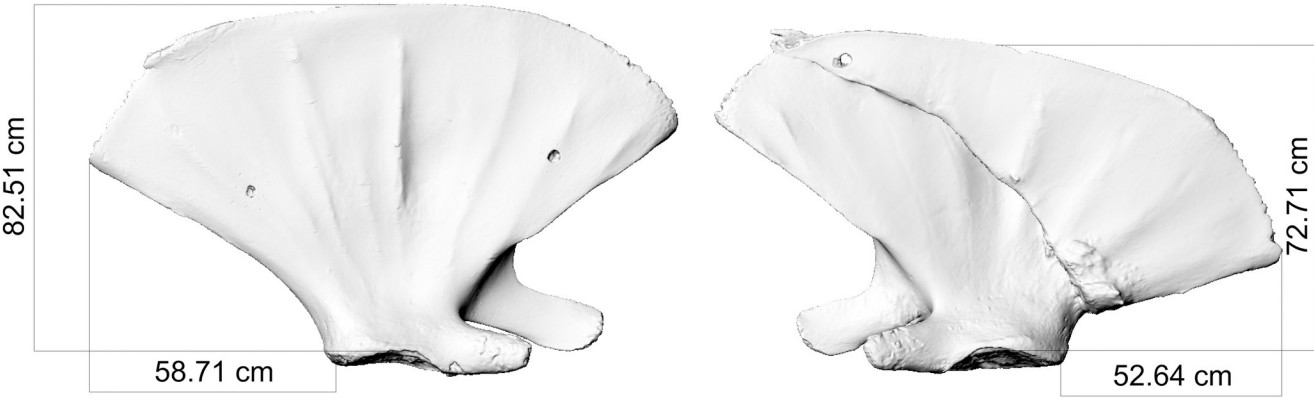

**Fig 5. Comparison of the unfractured left scapula and fractured right scapula of fin whale ZMH S 10261/10616 in external view.** From the measurements a 10 cm offset is evident.

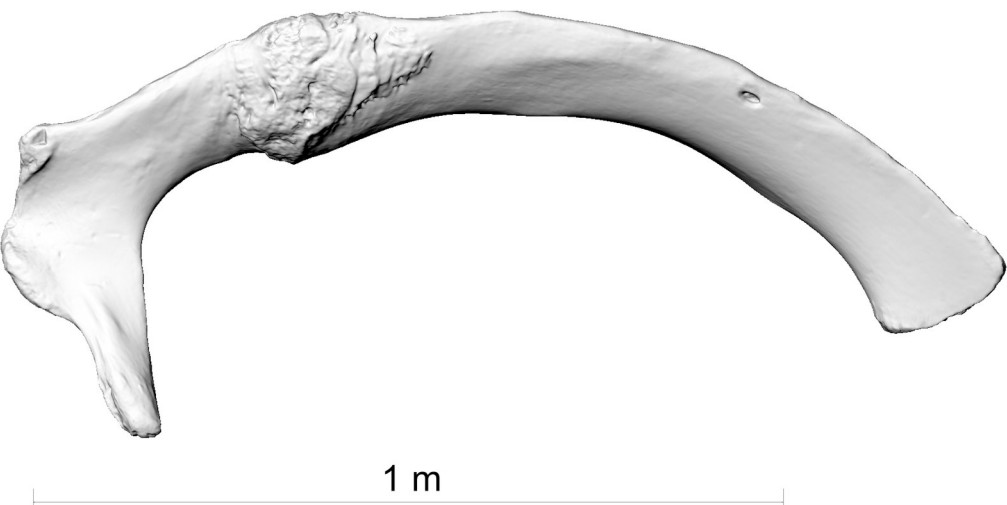

**Fig 6. The first right rib of the fin whale skeleton ZMH S 10261/10616 in caudal view.**

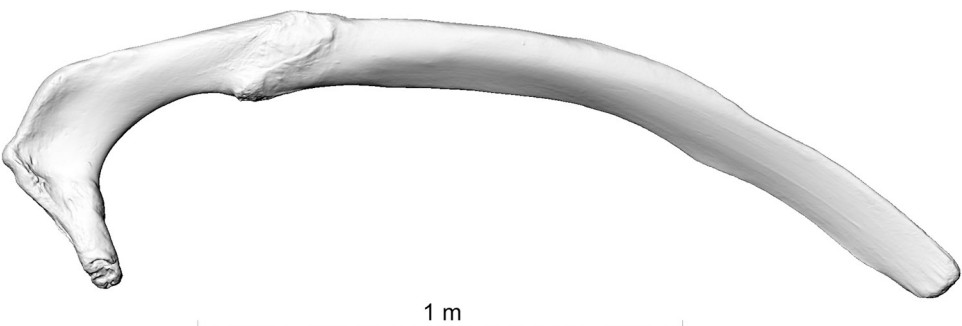

**Fig 7. The second right rib of the fin whale skeleton ZMH S 10261/10616 in caudal view.**

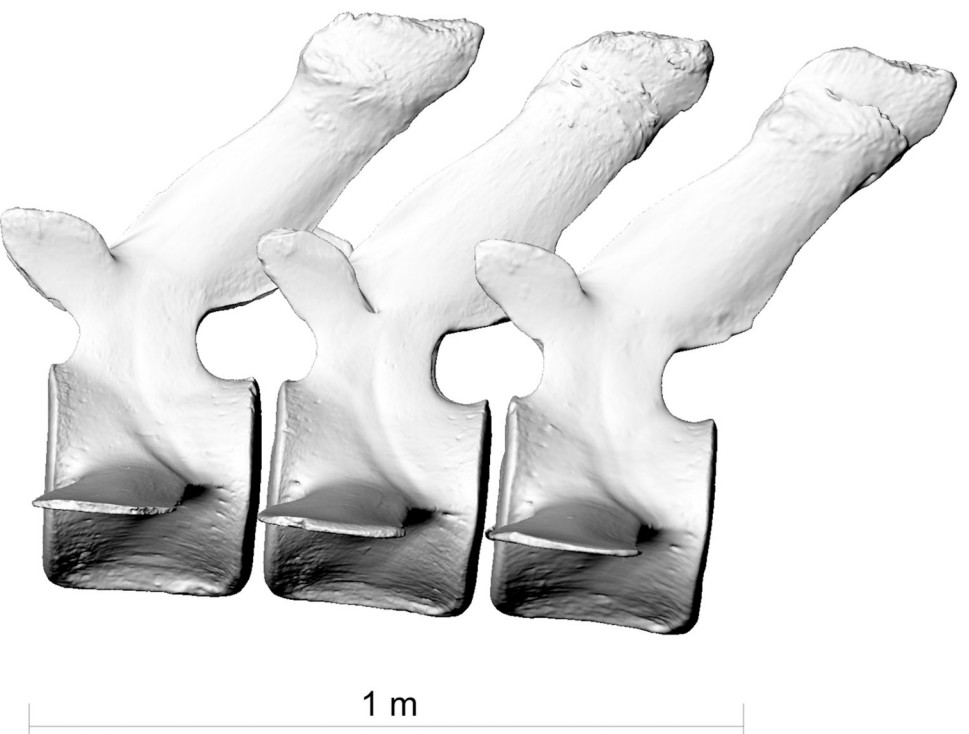

**Fig 8. Lumbar vertebrae L8, L9 and L10 of the fin whale ZMH S 10261/10616 in left lateral view.** The processus spinosi show healed fractures in a line descending from front to back.

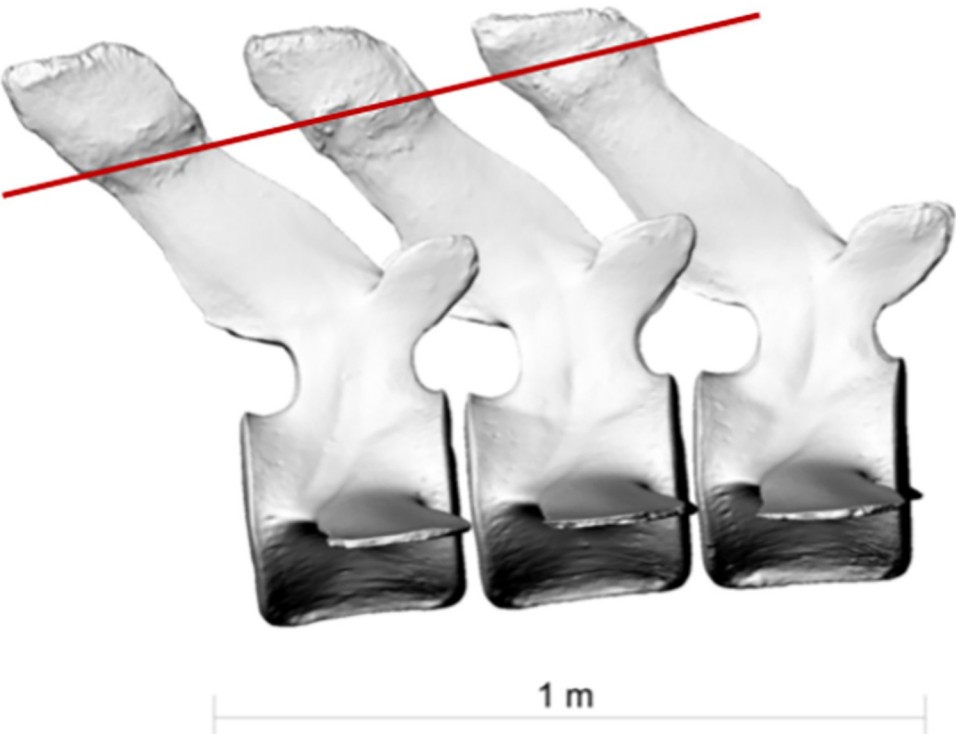

**Fig 9. Lateral view of lumbar vertrebrae L8, L9 and L10 of the fin whale ZMH S 10261/10616 in right lateral view.** Red line indicating the line of excessive bending moment imposed on the whale's spine likely causing the fractures on the spinal processes.

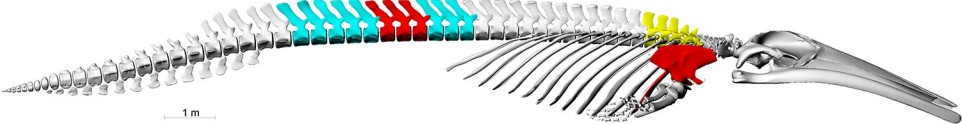

**Fig 10. Skeleton of the fin whale ZMH S 10261/10616.** Lumbar vertebrae anterior and posterior of the fractured vertebrae with deformations of their processus spinosi (marked in blue), likely the result of a post traumatic posture damage. Vertebrae with arthrosis are marked in yellow. Bones showing healed fractures are marked in red.

Barco [21] describe comparable injuries, a broken mandible and cranium of a North Atlantic right whale.

We therefore reconstructed a possible ship strike scenario explaining the observed pattern of injuries and deformations in the skeleton. In this scenario, the fin whale would have been hit laterally by the bow of a ship, which caused fractures of the scapula and the ribs (Figs 14 and 15). The localities of the fractures and the straight break of the scapula, suggest collision

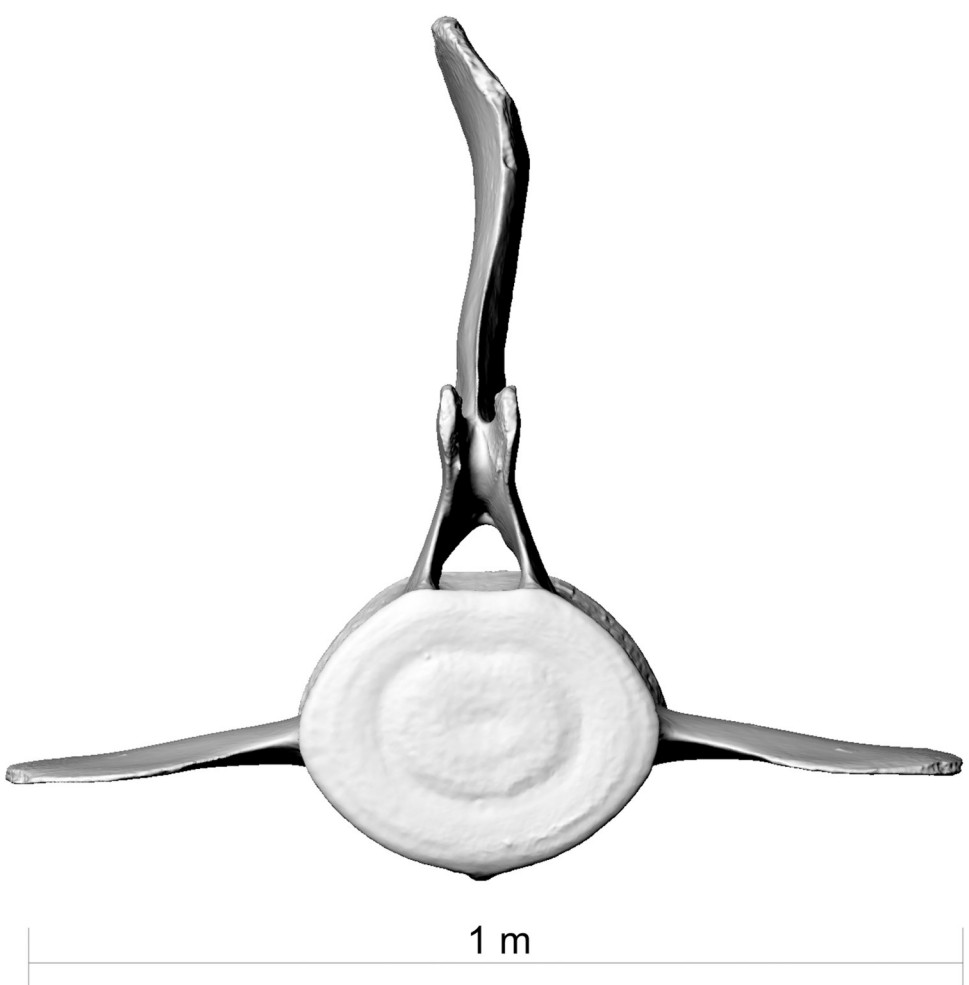

**Fig 11. Lumbar vertebra 14 of fin whale ZMH S 10261/10616 in posterior view.** Note strong left lateral bend in processus spinosus, likely the result of a post traumatic posture damage.

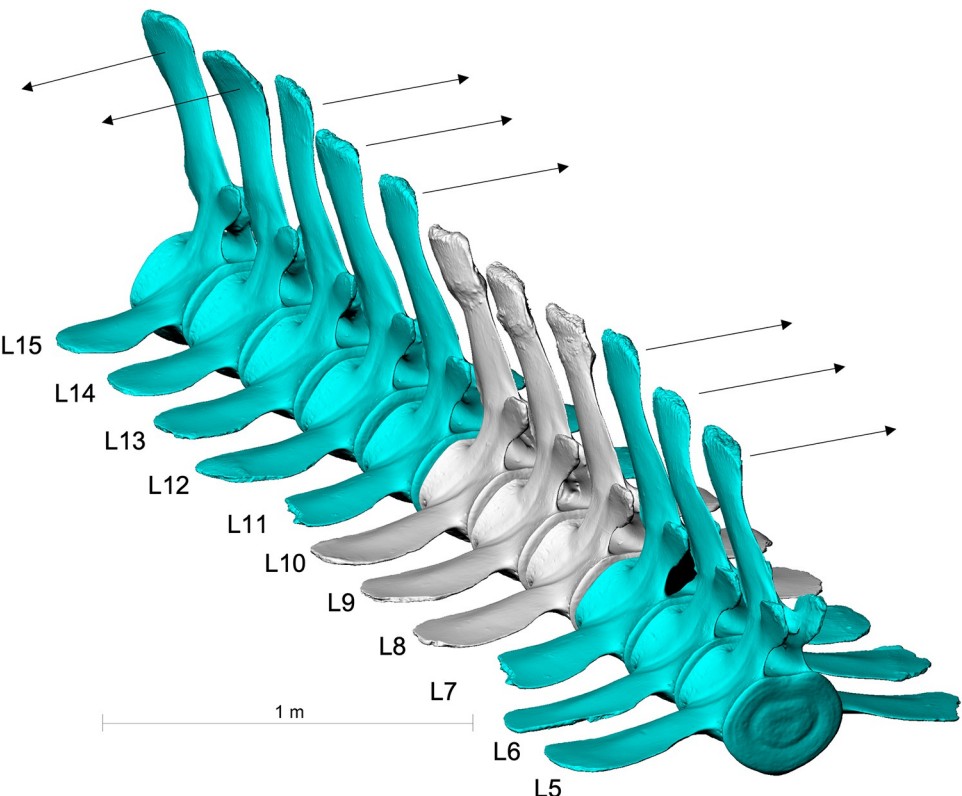

**Fig 12. Deformed vertebrae of fin whale ZMH S 10261/10616.** L5-L7 and L11-L13 surrounding the broken vertebrae L8-L10 are bended towards the left, L14-L15 are bended to the right.

with a vessel featuring a vertical steven or straight edged bow as commonly seen in military vessels built in the early decades of the 20th century, before the bulbous bow became more common in the early 1950s [22]. Whether this bow was a straight and narrow construction, sharp enough to cause penetrating trauma to soft tissues and bone, or a broader construction that resulted in blunt but localized trauma that snapped the upper part of the scapula blade via dislocation remains unclear. The breaks in a cranium and mandible of a Northern right whale reported by Campbell-Malone, Barco [21] resulted from blunt trauma, as evidenced by the bruised but not lacerated soft tissues on the carcass. Still, the breaks are straight and look as if they had been caused by a sharp blade impacting the bone directly. In the absence of histological data on either soft tissue we cannot determine the exact type of impact that caused the broken scapula in ZMH S 10261/10616.

The two broken ribs (Figs 6 and 7) were probably fractured by this same impact, when the upper part of the scapula was pushed deep into the body of the whale. The rib fractures are located directly medially to the scapula fracture line (Figs 1, 14 and 16).

Arthrosis observed in the thoracic vertebrae T2-T6, which are located directly above the scapula, may be a long-term consequence of a joint contusion caused by the same impact. Joint trauma can induce a contusive chondropathy responsible for post traumatic degenerative joint disease [23].

This scenario however does not yet explain the fractures in the processus spinosi of lumbar vertebrae L8-10. These could have either resulted from another traumatic incident, making the

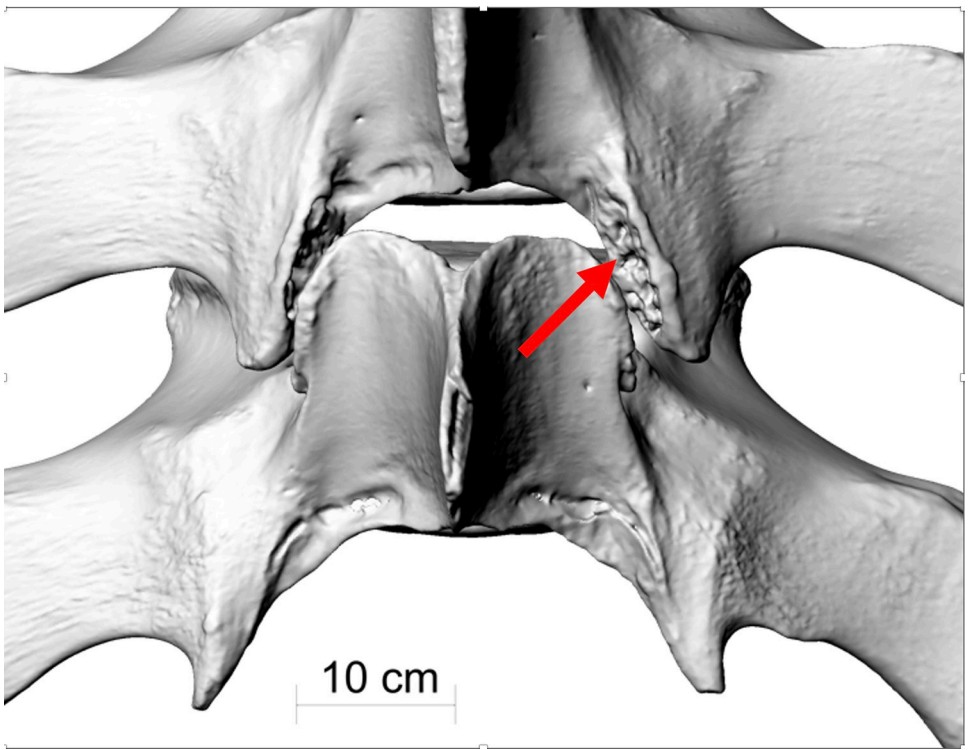

**Fig 13. The contact between thoracic vertebrae T3 and T4 of fin whale ZMH S 10261/10616 in dorsal view, showing evidence of arthrosis in the zygapophysial joints (red arrow).**

fin whale a survivor of two ship strikes. However, the similar healing stage of the injuries points to infliction at the same point in time rather than two independent collision events. The injuries to the lumbar vertebrae likely stem from a second collision with the same ship within the same incident. In this scenario, the whale would have been forced down by the initial impact or may have attempted to dive away from the ship. This motion and the continued forward motion of the ship may have brought the lumbar region of the whale into contact with the keel (Figs 17 and 18) or, less likely, the propeller of the ship resulting in fractures in the processus spinosi of lumbar vertebrae L8-10. However, fractures of the processus of the vertebrae cannot be easily attributed to external trauma, since these bones are protected by a thick layer of blubber and muscle [24–26]. Stress fractures of the processus may however be caused by excessive tension on the osteo-articular tendon structures, because the most potent locomotor muscles of cetaceans have their origins in these processus, especially in the thoracic region and lumbar region [25]. Therefore, an alternative hypothesis may be that, at the moment of the collision, the whale was forcibly flexed by (or around) the ship immediately after the impact. This excessive bending moment imposed on the whale's spine may have caused the fractures on the spinal processus of the lumbar vertebrae L8-10 [27].

As a consequence of these severe injuries, it is likely that the fin whale had to maintain a relieving posture for a extended period of time. Deformations in the vertebrae posterior and anterior of the fractured vertebrae may be the result of a prolonged posture damage. Likely, the fractured scapula impaired the movement particularly of the right flipper. In addition, broken ribs and haemorrhaging probably caused pain and may have caused the whale to attain a

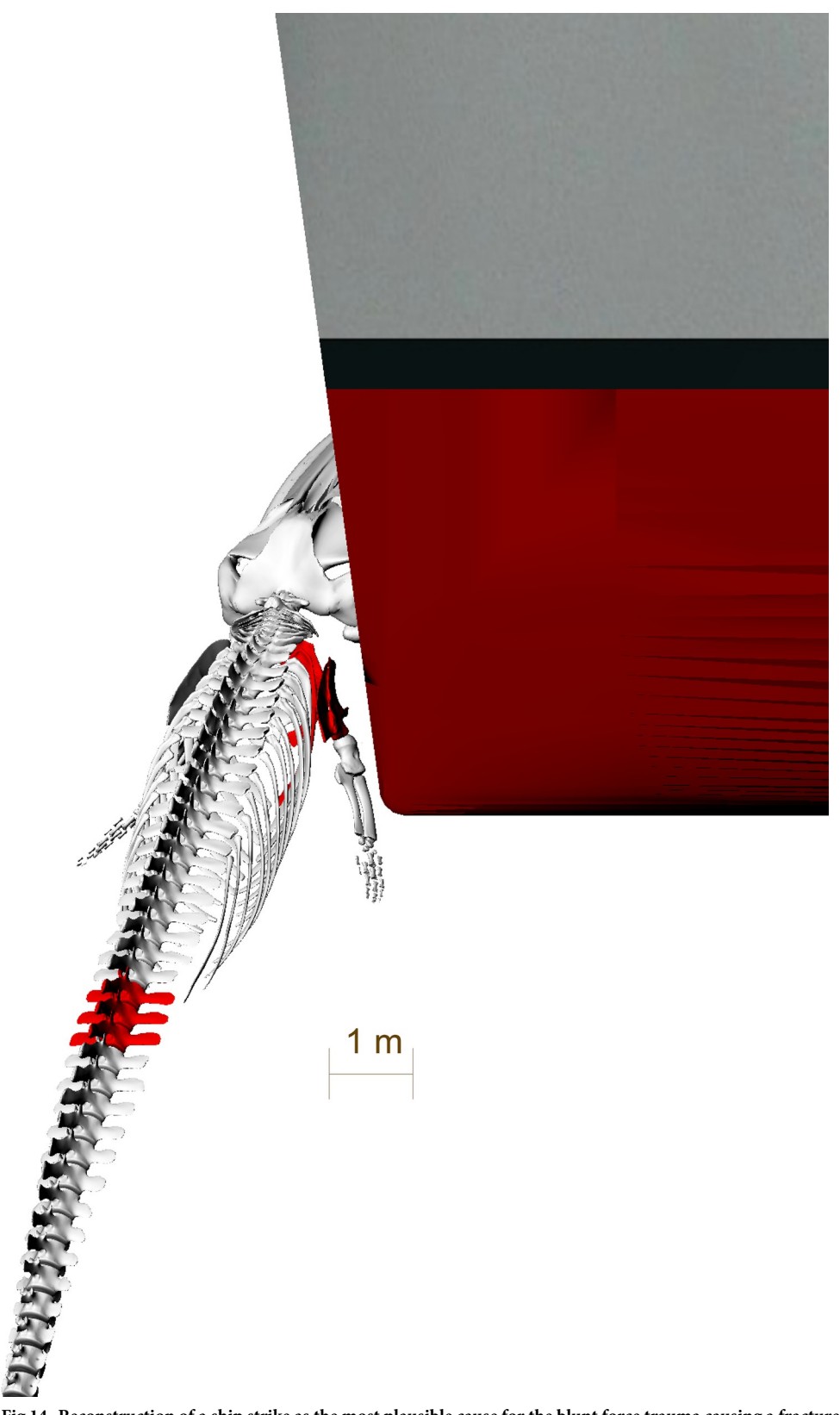

**Fig 14. Reconstruction of a ship strike as the most plausible cause for the blunt force trauma causing a fracture of the scapula and the ribs. The fin whale is hit laterally by a ship with a straight bow.** Generated in Rhinoceros 5.0.

Ship model represents HMS Ajax (http://www.naval-history.net/xGM-Chrono-06CL-Ajax.htm), a ship of the time period estimated for the collision. Model taken from: http://tf3dm.com/3d-model/british-light-cruiser-hms-ajax1939-39659.html.

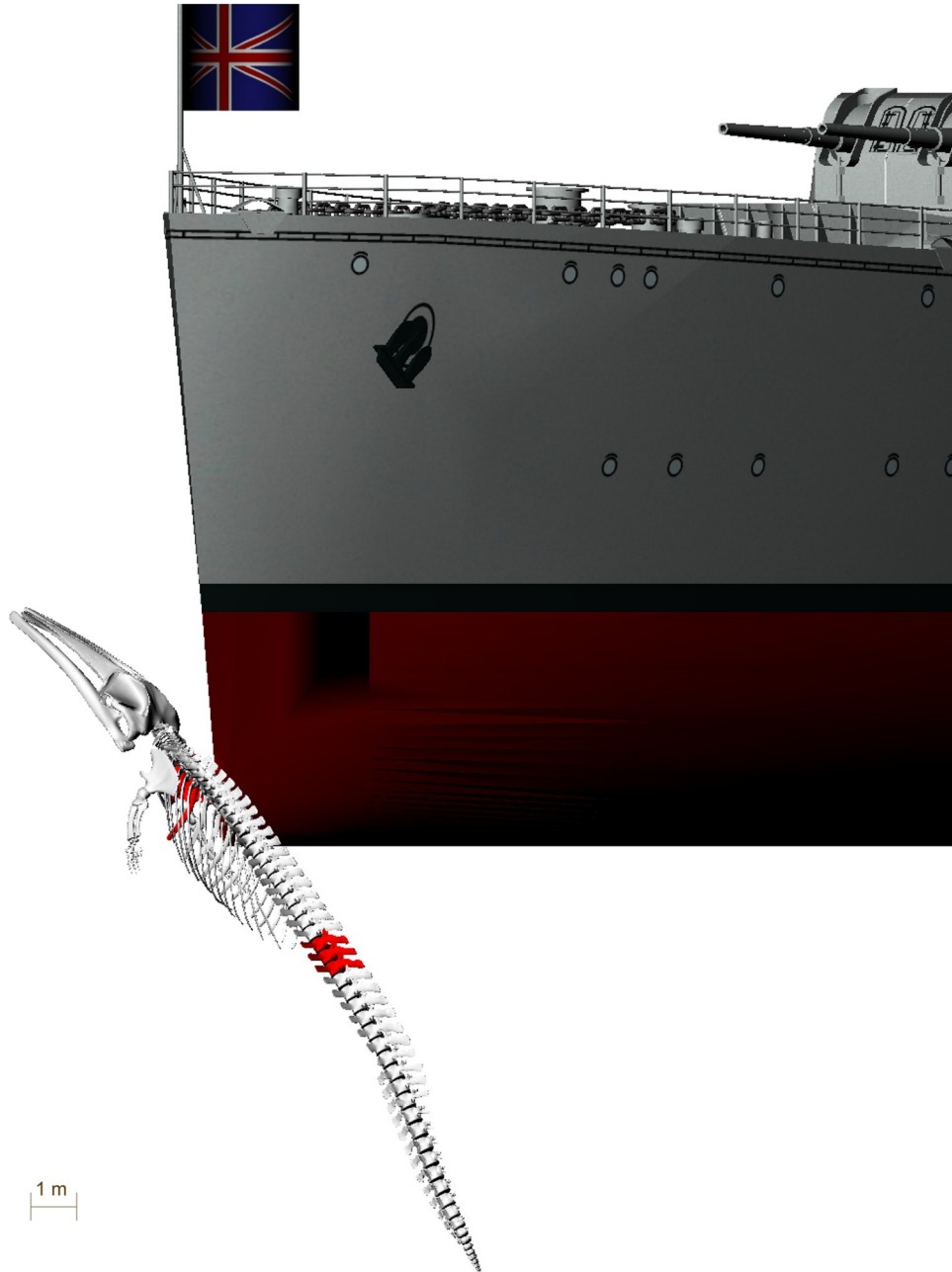

**Fig 15. Front *v*iew of the reconstruction of the ship strike as the most plausible cause for the blunt force trauma causing fractures to the scapula and ribs.** Generated in Rhinoceros 5.0. Ship model represents HMS Ajax, a ship of the time period estimated for the collision. Model taken from: http://tf3dm.com/3d-model/british-light-cruiser-hms-ajax1939-39659.html.

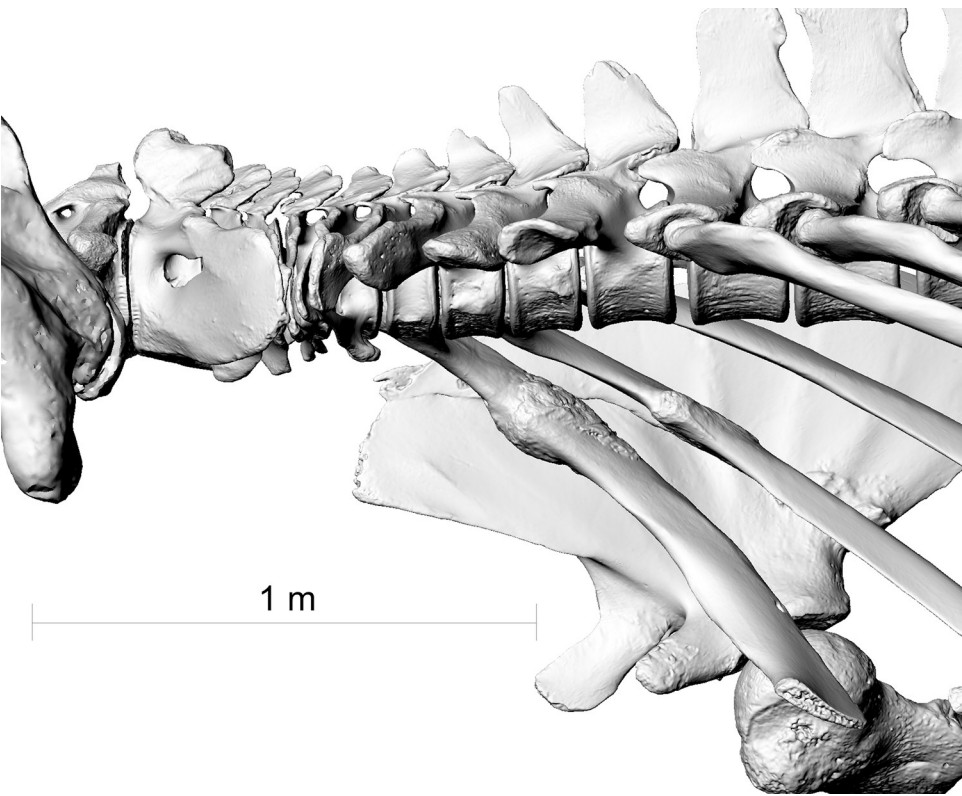

**Fig 16. Fractured scapula and ribs of the fin whale ZMH S 10261/10616.** Note: The rib fractures are located directly medially to the scapula fracture line.

relieving posture, sparing the right side and dominantly using the left. This may have led to an uneven muscular strain onto spinal processus of the lumbar vertebrae in which the locomotor muscles have their origin [25] particularly of the vertebrae around the broken vertebrae in order to relieve strain on these, causing a bending over the prolonged time of the relieving posture.

Assuming, that the fin whale was hit by a ship, this would be the first detailed documentation of a historical ship strike in a Southern Hemisphere fin whale. Ship strikes evolved as a major threat for whales over the last century [28] as the size, speed and quantity of ships increased [29,30]. Fin whales are one of the species most frequently affected by ship strikes [30], with ship strike rates giving reason for concern in areas such as the Mediterranean [31,32]. In our case, the fin whale survived the ship strike despite the severity of the injuries, as indicated by the fact that the fractures had completely healed by the time it was killed by the whaler. The relatively low rate of remodeling in cetacean bone [33] and the necessity for continued use of the extremity suggests, that the time of the collision must have been several years before the fin whale was shot in 1952, placing the ship collision in the 1940s.

The consequences of the injury for the individual can only be hypothesized. It is likely that the whale's movements were considerably impaired, as suggested by the evidence for a relieving posture, potentially affecting feeding efficiency, energy budgets and swim speed. It remains unknown, if the pre-condition made the individual an easier target for the whalers than other individuals.

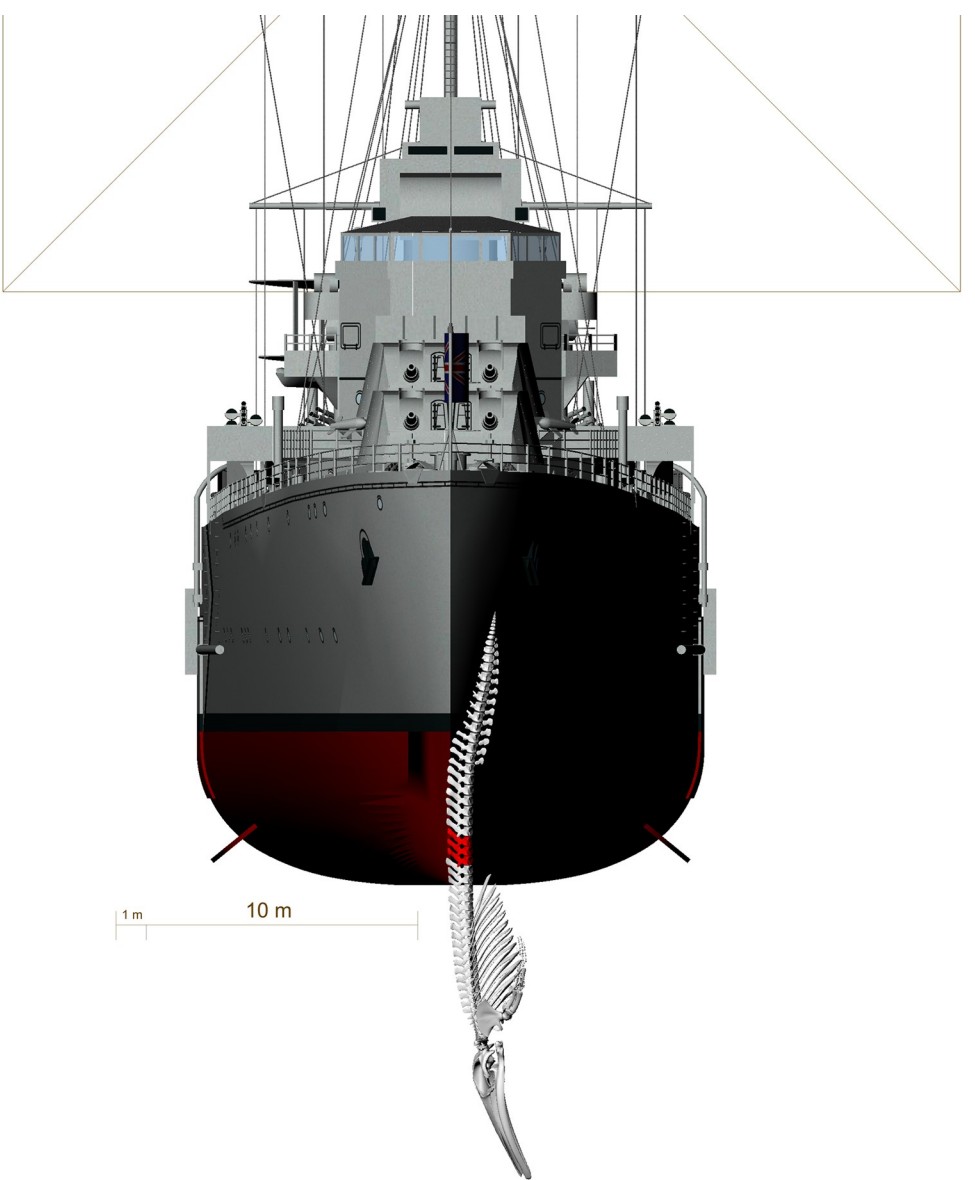

**Fig 17. Front view of the reconstruction of a possible second collision as a cause for the blunt force trauma causing fractures to the lumbar vertebrae L8-L10.** Generated in Rhinoceros 5.0. Ship model represents HMS Ajax, a ship of the time period estimated for the collision. Model taken from: http://tf3dm.com/3d-model/british-light-cruiser-hms-ajax1939-39659.html.

While sharp trauma caused by propellers and rudders during ship strikes usually provides obvious external evidence of injury, blunt trauma like in the case of our fin whale typically leaves little external evidence of the injury, even in the most severe cases, because of the thickness of soft tissue and the dark pigment of much of the epidermis, which tends to obscure evidence of swelling and bruising [21]. Even severe injuries like the one reported here may go unnoticed in visual observations of live animals as well as stranded carcasses that are not necropsied. Therefore, cases like the fin whale in this study may contribute to the overall number of ship strikes, but remain undetected.

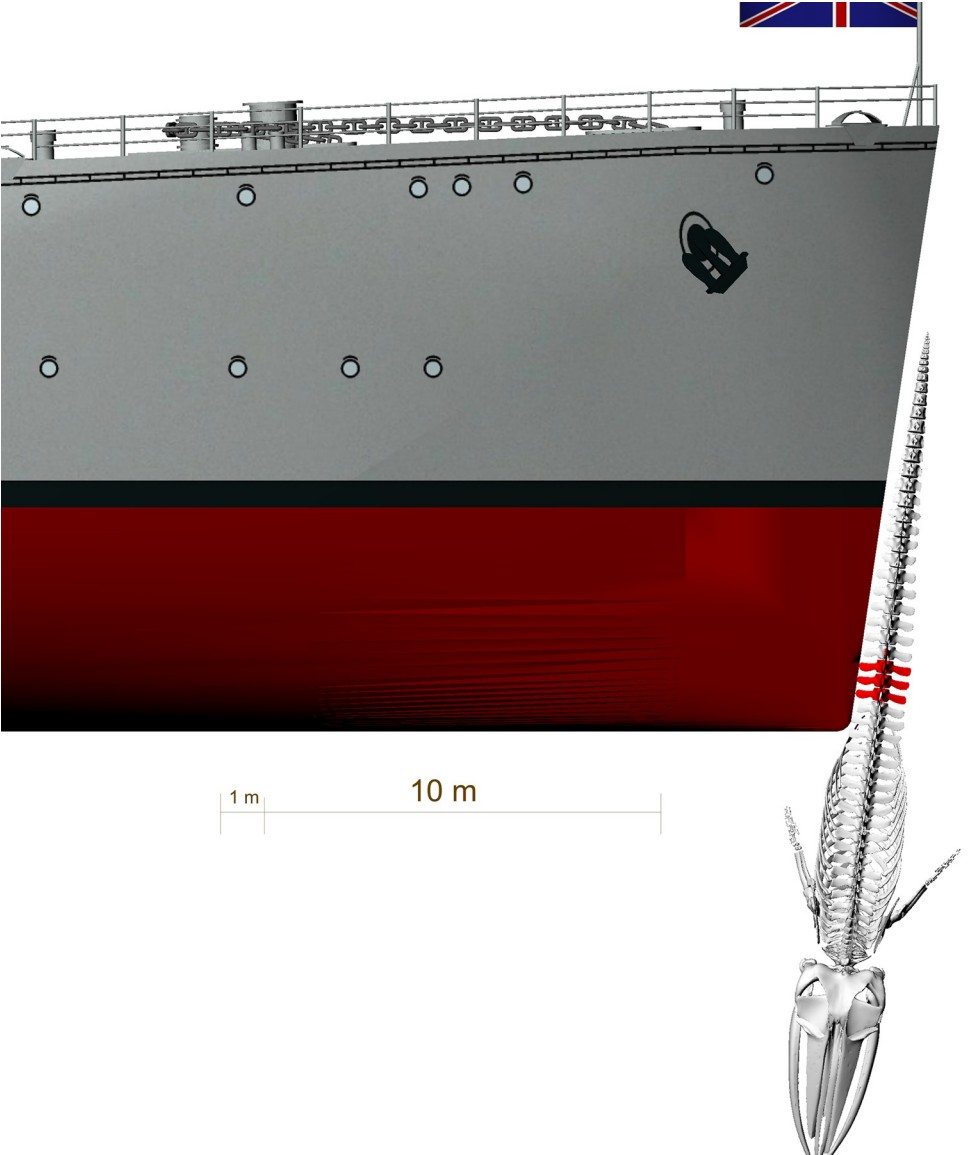

**Fig 18. Side view of the reconstruction of a possible second collision as a cause for the blunt force trauma causing fractures to the lumbar vertebrae L8-L10.** Generated in Rhinoceros 5.0. Ship model represents HMS Ajax, a ship of the time period estimated for the collision. Model taken from: http://tf3dm.com/3d-model/british-light-cruiser-hms-ajax1939-39659.html.

This study highlights the value of museum collections for research on threatened species and insights into the life history of individuals.

## Acknowledgments

We would like to thank Matthias Preuß and Steffen Lässle, taxidermists at the Zoological Museum of Hamburg, for providing their expertise during the digital assembly of the fin whale skeleton.

## Author Contributions

**Conceptualization:** Matthias Glaubrecht, Thomas M. Kaiser.

**Data curation:** Matthias Glaubrecht, Thomas M. Kaiser.

**Formal analysis:** Hannah Viola Daume, Thomas M. Kaiser.

**Funding acquisition:** Thomas M. Kaiser.

**Investigation:** Hannah Viola Daume, Heinrich Mallison, Thomas M. Kaiser.

**Methodology:** Heinrich Mallison, Thomas M. Kaiser.

**Project administration:** Thomas M. Kaiser.

**Resources:** Heinrich Mallison, Thomas M. Kaiser.

**Software:** Thomas M. Kaiser.

**Supervision:** Helena Herr, Matthias Glaubrecht, Thomas M. Kaiser.

**Validation:** Helena Herr, Thomas M. Kaiser.

**Visualization:** Hannah Viola Daume.

**Writing – original draft:** Hannah Viola Daume, Helena Herr.

**Writing – review & editing:** Hannah Viola Daume, Helena Herr, Heinrich Mallison, Matthias Glaubrecht, Thomas M. Kaiser.

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
