## [Decision Letter · Decision Letter 0]

16 Nov 2022

PONE-D-22-27969Osteo-pathological analysis provides evidence for a survived historical ship strike in a Southern Hemisphere fin whale (Balaenoptera physalus)PLOS ONE

Dear Dr. Daume,

Thank you for submitting your manuscript to PLOS ONE. After careful consideration, we feel that it has merit but does not fully meet PLOS ONE’s publication criteria as it currently stands. Therefore, we invite you to submit a revised version of the manuscript that addresses the points raised during the review process.

Your manuscript has been reviewed by two highly qualified reviewers. Both of them agree (and I tend to agree with them) that this is a very well-written and interesting study that is worth to be published in PLOS ONE. Nevertheless, both reviewers have suggested some minor changes that will help to improve the paper.

When revising your manuscript, please consider all issues mentioned in the reviewers' comments carefully: please, outline every change made in response to their comments and provide suitable rebuttals for any comments not addressed.

Please, note that your revised submission may need to be re-reviewed.

We look forward to receiving your revised manuscript.

Kind regards,

Olga Spekker, Ph.D.

Academic Editor

PLOS ONE

Journal Requirements:

Reviewers' comments:

Reviewer's Responses to Questions

**Comments to the Author**

1. Is the manuscript technically sound, and do the data support the conclusions?

Reviewer #1: Yes

Reviewer #2: Partly

2. Has the statistical analysis been performed appropriately and rigorously? 

Reviewer #1: N/A

Reviewer #2: N/A

3. Have the authors made all data underlying the findings in their manuscript fully available?

Reviewer #1: Yes

Reviewer #2: Yes

4. Is the manuscript presented in an intelligible fashion and written in standard English?

Reviewer #1: Yes

Reviewer #2: Yes

5. Review Comments to the Author

Reviewer #1: This is a compact and clear case study, providing results for basic research into cetacean pathology and its applied aspect in the welfare of whales. It also shows the increasing importance of "old fashoned" museum collections in research concerning threatened species which may be worth emphasizing in the text.

The article is clearly written, supported by high quality illustrations obtained through sophisticated imaging methods.

Perhaps the description of directions could be clarified (actually simplified) in the foollowing figures:

Line 109: Figure 2: ... in lateral view from the external side

Line 113: Figure 3: ... in lateral view from the internal side.

I find the use of "lateral" confusing here as I suspect that the "external" side is lateral and the "internal" side is medial. I presume the original meaning was a reference to the lateral view of the skeleton in general. If I misunderstood something, a brief explanation would help other readers.

Line 140-143: Figures 8-9: ... left dorsolateral view

I think this is simply the lateral view as I do not see the vertebra being shown from "slightly above".

There seems to be superfluous wording here:

Line 147: Several lumbar vertebrae anterior (L5-L7) and posterior (L11-L15), [posterior and anterior of the...]

The text in square brackets could be deleted.

A reference could be added to a concrete parameters of HMS Ajax here:

Line 203: represents HMS Ajax, a ship of the time period estimated for the collision.

Although in the discussion proper references are made to a variety of accidents, it would be worth devoting a sentence in the text on how many similar traumatic cases are known. My impression is that this observation is quite unique, but it would be worth placing it in a slightly broader, concrete perspective.

Reviewer #2: The submitted manuscript [PONE-D-22-27969] concerns the recognition of traumatic pathologies on the skeleton of a whale (Balaenoptera physalus) and the reconstruction of the traumatic event. The fractures were probably caused, according to the authors, by a collision with a ship in the 1940s. This whale was killed by a whaler in the South Atlantic in 1952 and its skeleton (ZMH S 10261/10616) belongs to the Hamburg Zoological Museum, which has been exhibiting it since 2015.

The authors used 2 types of 3D surface imaging: structured light scanning and photogrammetry. The interest of the 3D surface imaging used in this article lies mainly in the creation of a virtual model of the whale and the reconstruction of the traumatic event. Indeed, it does not help in retrospective diagnosis, as it does not offer more significant information than a set of good photographic images. A X-ray study of the damaged bones, for example, would have been interesting to visualize the internal bone structures, bringing additional information to the diagnosis of these consolidated fractures.

The adjustment of the different elements of the skeleton to propose a general 3D reconstruction of the whale, carried out by the authors according to their experience is interesting and useful to better understand both the location of the lesions and the history of the trauma. It seems to me to be well done, even if I am not a specialist in the anatomy of the whale skeleton.

My main comments are about some of the skeletal lesions and the interpretations:

- although there are no supporting radiographs, I agree with the authors on the diagnosis of healed fractures of the right scapula, the right ribs of ranks 1 and 2, and the lumbar vertebrae of ranks 8 to 10. The signs visible on the external surface of the 3D reconstructions of these bones are quite characteristic of traumatic conditions and do not require a differential diagnosis.

- the authors observed deformations of the spinous processes of the lumbar vertebrae located on either side of the fractured vertebrae, i.e. lumbar vertebrae of ranks 5 to 7 and 8 to 10. These deformations are interpreted by the authors as a condition due to posture. I would like the authors to better argue this interpretation : would they not be consolidated fractures of the spinous processes ? Indeed in figure 11, the orientation of the spinous process is very clear but could as well result from direct impact with the bow of the ship and not secondary to the animal’s posture due to pain.

- similarly, the authors interpret the origin of the degenerative joint disease observed in the upper thoracic vertebrae (T2 to T6) as postural in origin. It would be interesting to illustrate these osteoarthritic vertebrae on the figures of the reconstructions of the animal's bone lesions (figures 1, 10, 13 and 14). This would make it possible to discuss an alternative hypothesis: the degenerative phenomenon observed on the posterior vertebral joints at the thoracic level could be due, not to a postural cause, but to a joint contusion linked to the same trauma responsible for the fractures of the scapula and the ribs. Indeed, any joint trauma can induce a « contusive chondropathy » responsible for post-traumatic degenerative joint disease.

- the authors mention in lines 194-195, that in the absence of bone histology data, it is difficult to determine the type of impact that caused the scapula fracture. If the authors believe that bone histology would differentiate between a blunt and a penetrating agent, I would like them to support their assertion here.

- the authors propose an alternative hypothesis to the presence of two fracture sites on the whale at the scapula and the lumbar vertebrae (lines 224 to 227): indirect trauma by excessive flexion of the spine. I am not convinced by this hypothesis. The interpretation of a first impact on the whale's scapula with the ship and then a second impact when the whale swung its tail to dive seems more plausible.

6. PLOS authors have the option to publish the peer review history of their article (what does this mean?). If published, this will include your full peer review and any attached files.

Reviewer #1: **Yes: **László Bartosiewicz

Reviewer #2: No

---

## [Decision Letter · Decision Letter 1]

8 Jan 2023

PONE-D-22-27969R1Osteo-pathological analysis provides evidence for a survived historical ship strike in a Southern Hemisphere fin whale (Balaenoptera physalus)PLOS ONE

Dear Dr. Daume,

Thank you for submitting your manuscript to PLOS ONE. After careful consideration, we feel that it has merit but does not fully meet PLOS ONE’s publication criteria as it currently stands. Therefore, we invite you to submit a revised version of the manuscript that addresses the points raised during the review process.

Comments from the Academic Editor:

Dear Dr. Daume,

We appreciate you submitting your manuscript to PLOS ONE and thank you for giving us the opportunity to consider your work.

I have completed my evaluation of your revised manuscript, which has been reviewed by two highly qualified reviewers all of whom agree it is worth to be published in PLOS ONE. Nevertheless, I kindly ask you to review your in-text citations and reference list, as they do not always follow the guidelines of PLOS ONE (https://journals.plos.org/plosone/s/submission-guidelines#loc-references). The references should be numbered in the order that they appear in the text. In the text, the reference number should be cited in square brackets. As for the formatting of the references, PLOS ONE uses the so-called „Vancouver style”. (I copied below the relevant parts of the guide for authors.)

„References are listed at the end of the manuscript and numbered in the order that they appear in the text. In the text, cite the reference number in square brackets (e.g., “We used the techniques developed by our colleagues [19] to analyze the data”). PLOS uses the numbered citation (citation-sequence) method and first six authors, et al.”

„PLOS uses the reference style outlined by the International Committee of Medical Journal Editors (ICMJE), also referred to as the “Vancouver” style. Example formats are listed below. Additional examples are in the ICMJE sample references…”

In addition to the above, I kindly ask you to follow the guidelines of PLOS ONE regarding the figure captions (e.g., **Fig 1.**
**This is the Fig 1 Title.** This is the Fig 1 legend) and in-text citation of figures (e.g., Fig 1, Figs 1 and 2, Figs 1, 2, and 3) (https://storage.googleapis.com/plos-published-prod/9cba/PLOS%20Manuscript%20Body%20Formatting%20Guidelines.pdf?X-Goog-Algorithm=GOOG4-RSA-SHA256&X-Goog-Credential=wombat-sa%40plos-prod.iam.gserviceaccount.com%2F20230107%2Fauto%2Fstorage%2Fgoog4_request&X-Goog-Date=20230107T075032Z&X-Goog-Expires=86400&X-Goog-SignedHeaders=host&X-Goog-Signature=b73dd06d9f0d74651d636752c03e6d4ac51f9bfac573ffa1f75af7211c142093e3e4458c4912e230fe884d2bcc7f1d508da875b4e3e4268966a094a597195b9aabdbacf099f08920ca1b6837a31d4826aa8eadc4e9a0f577f6a79f25d9be460c6d8eb0af67ae695413a64b21a9f16fe2e1fe26070ca34a1fe3a267a8cc592a56b8f8edf1a857471c0f78a9e17d58fdb7c87900f958a02a78b462f9e91c55a4edeeab68fa45cb5d6f762232530827970a78c4eef21977ce02648fc67a4599283f28986c328bb6853cf2f4364e0db00c6730a662e9fe0eb64fa64ecfc496e7b90dab5880993adf80904c9ce43e620b2372ea692432a2e579a8ea1b04228be72ad1).

Furthermore, figure files should be uploaded separately, instead of being included in the main manuscript file. (I copied below the relevant part of the guide for authors.)

Do not include figures in the main manuscript file. Each figure must be prepared and submitted as an individual file.

Cite figures in ascending numeric order at first appearance in the manuscript file.

Figure captions must be inserted in the text of the manuscript, immediately following the paragraph in which the figure is first cited (read order). Do not include captions as part of the figure files themselves or submit them in a separate document.

At a minimum, include the following in your figure captions:

A figure label with Arabic numerals, and “Figure” abbreviated to “Fig” (e.g. Fig 1, Fig 2, Fig 3, etc). Match the label of your figure with the name of the file uploaded at submission (e.g. a figure citation of “Fig 1” must refer to a figure file named “Fig1.tif”).A concise, descriptive title

The caption may also include a legend as needed.”

Finally, please find below some minor comments:

1) There are inconsistencies in the writing of some terms: e.g., life history or life-history; 1950ies/1940ies or 1950s/1940s; post traumatic or post-traumatic; ship-strike or ship strike; 3D models, 3D-surface or 3-D model - please, revise the text and eliminate these inconsistencies;

2) There are some typos or grammatical issues: e.g., line 76 – „or even not not suitable”; line 92 – „digital digiti”; line 166 – instead of its, their or the should be used; line 242 – „locomotors muscles” should be changed to locomotor muscles; lines 246–247 – „on the vertebra’s spinal processus” should be rephrased; lines 269–270 – „as the ships size, speed and quantity” should be rephrased (e.g., as the size, speed and quantity of ships) - please, eliminate the aforementioned issues; and

3) Figures and their captions: in the caption of figure 2, it seems that a space is missing between the specimen number and from; the caption of figures 6 and 7 should be rephrased to unify and simplify the captions (“The first right rib of fin whale ZMH S 10261/ 10616 in caudal view”; and “The second right rib of fin whale ZMH S 10261/ 10616 in caudal view.”); in the caption of figures 15, 17, and 18, “view” should not be capitalised; in the caption of figure 2, “of” is in a different font type than the rest of the caption; in the caption of figure 5, maybe external should be used instead of lateral, as in figure 2; in the caption of figure 9, one of the reviewers already mentioned that “dorso” should be deleted; and in figure 10, the “green” seems to me more like blue, and maybe the scapula should not be highlighted in red, as in the legend, only the vertebrae are mentioned.

Based on the above, I invite you to resubmit your manuscript after amending it following the aforementioned comments and suggestions. Please, note that your revised submission may need to be re-reviewed.

PLOS ONE values your contribution and I look forward to receiving your revised manuscript.

Yours sincerely,

Dr. Olga Spekker

We look forward to receiving your revised manuscript.

Kind regards,

Olga Spekker, Ph.D.

Academic Editor

PLOS ONE

Journal Requirements:

Reviewers' comments:

Reviewer's Responses to Questions

**Comments to the Author**

1. If the authors have adequately addressed your comments raised in a previous round of review and you feel that this manuscript is now acceptable for publication, you may indicate that here to bypass the “Comments to the Author” section, enter your conflict of interest statement in the “Confidential to Editor” section, and submit your "Accept" recommendation.

Reviewer #1: (No Response)

Reviewer #2: All comments have been addressed

2. Is the manuscript technically sound, and do the data support the conclusions?

Reviewer #1: Partly

Reviewer #2: Yes

3. Has the statistical analysis been performed appropriately and rigorously? 

Reviewer #1: N/A

Reviewer #2: N/A

4. Have the authors made all data underlying the findings in their manuscript fully available?

Reviewer #1: Yes

Reviewer #2: Yes

5. Is the manuscript presented in an intelligible fashion and written in standard English?

Reviewer #1: Yes

Reviewer #2: Yes

6. Review Comments to the Author

Reviewer #1: (No Response)

Reviewer #2: The authors of the manuscript PONE-D-22-27969R1 have answered all the questions asked in my first review. They have taken into account all my remarks and suggestions, including the addition of a new figure (12) representing in 3D the concerned vertebrae, which clearly supports their interpretations.

This revised version seems to me now acceptable for publication in the journal Plos One.

7. PLOS authors have the option to publish the peer review history of their article (what does this mean?). If published, this will include your full peer review and any attached files.

Reviewer #1: No

Reviewer #2: **Yes: **Helene COQUEUGNIOT

---

## [Editor Report · Decision Letter 2]

20 Jan 2023

Osteo-pathological analysis provides evidence for a survived historical ship strike in a Southern Hemisphere fin whale (Balaenoptera physalus)

PONE-D-22-27969R2

Dear Dr. Daume,

We’re pleased to inform you that your manuscript has been judged scientifically suitable for publication and will be formally accepted for publication once it meets all outstanding technical requirements.

Kind regards,

Olga Spekker, Ph.D.

Academic Editor

PLOS ONE